# Cardiac Acetylation in Metabolic Diseases

**DOI:** 10.3390/biomedicines10081834

**Published:** 2022-07-29

**Authors:** Emilie Dubois-Deruy, Yara El Masri, Annie Turkieh, Philippe Amouyel, Florence Pinet, Jean-Sébastien Annicotte

**Affiliations:** Univ. Lille, Inserm, CHU Lille, Institut Pasteur de Lille, U1167-RID-AGE-Facteurs de Risque et Déterminants Moléculaires des Maladies Liées au Vieillissement, F-59000 Lille, France; yara.elmasri@pasteur-lille.fr (Y.E.M.); ani.turkieh@pasteur-lille.fr (A.T.); philippe.amouyel@pasteur-lille.fr (P.A.)

**Keywords:** acetylation, heart, obesity, diabetes, heart failure, enzymes

## Abstract

Lysine acetylation is a highly conserved mechanism that affects several biological processes such as cell growth, metabolism, enzymatic activity, subcellular localization of proteins, gene transcription or chromatin structure. This post-translational modification, mainly regulated by lysine acetyltransferase (KAT) and lysine deacetylase (KDAC) enzymes, can occur on histone or non-histone proteins. Several studies have demonstrated that dysregulated acetylation is involved in cardiac dysfunction, associated with metabolic disorder or heart failure. Since the prevalence of obesity, type 2 diabetes or heart failure rises and represents a major cause of cardiovascular morbidity and mortality worldwide, cardiac acetylation may constitute a crucial pathway that could contribute to disease development. In this review, we summarize the mechanisms involved in the regulation of cardiac acetylation and its roles in physiological conditions. In addition, we highlight the effects of cardiac acetylation in physiopathology, with a focus on obesity, type 2 diabetes and heart failure. This review sheds light on the major role of acetylation in cardiovascular diseases and emphasizes KATs and KDACs as potential therapeutic targets for heart failure.

## 1. Introduction

Obesity is a worldwide epidemic associated with several public health challenges including type 2 diabetes, hypertension, obstructive sleep apnea, dyslipidemia and cardiovascular disease, and notably heart failure. The prevalence of these disorders (e.g., obesity, type 2 diabetes and heart failure) is still increasing and heart failure remains the most common cause of cardiovascular morbidity and mortality in the world [1]. Interestingly, although heart failure could provoke systolic or diastolic cardiac dysfunction, obesity and type 2 diabetes are more often characterized by cardiac hypertrophy and diastolic dysfunction [2]. Moreover, diastolic heart failure remains one of the more challenging diseases to treat and, thus, a better understanding of the physio(patho)logical mechanisms involved in metabolic heart disease is necessary to propose new pharmacological therapies. In this context, epigenetic or protein post-translational modifications, including phosphorylation, methylation, O-GlcNAcylation, ubiquitylation, and acetylation, could represent interesting targets since they modulate gene or protein expression during metabolic diseases [3].

Protein acetylation is a highly conserved mechanism that was first described 60 years ago, in which an acetyl group is covalently attached to the ε-amino group of lysine residues [4]. This modification induces important changes to the protein structure at its lysine residue, by altering its charge status and adding an extra structural moiety [5]. These post-translational modifications can regulate protein–protein interactions, stability and function. Acetylation was first described to regulate chromatin structure and histone activity, associated to epigenetic-mediated gene regulation [4]. Since then, several proteomic analyses enabled to identify more than a thousand acetylation sites in both histone and non-histone proteins [6,7,8], and, interestingly, revealed that the subcellular distribution of acetylation is tissue dependent [9]. Focusing on cardiovascular and metabolic diseases, Table 1 indicates some of the most common proteins modified by cardiac acetylation.

## 2. Cardiac Acetylation

### 2.1. Regulation of Lysine Acetylation

Lysine acetylation is a reversible mechanism that is regulated by the dynamic actions of lysine acetyltransferase (KATs) and lysine deacetylase (KDACs) enzymes [43]. There are three categories of KATs: the MYST family, the Gcn-5-related N-acetyltransferases (GNATs) and the E1A-associated protein of 300 kDa/CREB binding protein (p300/CBP) family [44]. KDACs are broadly categorized into four classes based on function and DNA sequence similarity. Class I (reduced potassium dependency 3 family), II (histone deacetylase 1 family) and IV (HDAC11) are considered as classical KDACs with zinc-dependent active sites, whereas class III enzymes are a family of silent information regulator 2-like nicotinamide adenine dinucleotide (NAD^+^)-dependent deacetylases/mono-ADP-ribosyl transferases known as sirtuins 1 to 7 [44]. Interestingly, the level of expression of KATs and KDACs follows a specific tissue distribution, suggesting specific functions in these different organs [45]. For example, in physiological conditions, KATs are weakly expressed in vascular tissue, except for the lysine acetyltransferase KAT2B, which is highly expressed [45]. Conversely, a database mining approach analyzing the expression profile of 164 enzymes in human and murine tissues highlighted heart as tissue in which KATs and KDACs are in the highest varieties [45]. In addition to tissue-specific expression, both KATs and KDACs have precise subcellular localizations related to their specific targets (Figure 1) and can regulate each other. For example, cardiomyocyte overexpression of SIRT6 decreases P300 levels, where SIRT6 promotes ubiquitination and degradation of P300 [46]. Moreover, cardiomyocyte overexpression of SIRT3 increases SIRT1 expression [27].

In addition, cofactors play an essential role to modulate acetylation. Several short-chain acyl-CoA products, such as acetyl-coenzyme A (acetyl-CoA), are important metabolite intermediates generated during catabolism of various energy fuels that modulate cardiac acetylation. Indeed, acetyl-CoA acts as a substrate for KATs. Moreover, auto-acetylation of mitochondrial proteins can occur efficiently at the pH of the matrix (pH ≥ 7.5) in actively respiring mitochondria [9]. On the other hand, levels of NAD^+^, an important cofactor for sirtuins, may impact protein acetylation [5]. Indeed, NAD^+^ levels, synthesized from metabolic pathways, connect their enzymatic activity to metabolism and provide a metabolic link between the sirtuins activity, energy homeostasis and stress responses.

### 2.2. Physiological Roles of Lysine Acetylation

Lysine acetylation can modulate cell growth, metabolism, protein–protein interactions, protein stability, subcellular localization, gene transcription, chromatin structure or enzymatic activity [6,7]. This post-translational modification plays an essential role in cardiac physiology (Figure 2).

#### 2.2.1. Heart Development and Cardiac Ageing

Acetylation can regulate embryonic heart development and cardiac progenitor fate [47,48]. TBX5, a T-box family transcription factor, is involved in heart and forelimb development and is acetylated by KAT2A and KAT2B at lysine 339, leading to stimulate its transcriptional activity [18]. Knockdown of *KAT2A* and *KAT2B* in zebrafish severely impairs heart development and induces pericardial oedema by inhibiting TBX5 [18]. Besides KATs, the lysine deacetylase enzymes HDAC4 and HDAC5 are also able to interact with TBX5 to exert a repressive role on cardiac genes transcription through deacetylation [19]. Furthermore, TBX1, another T-box family transcription factor expressed in cardiac progenitors, represses myocyte enhancer factor 2c *(Mef2c)* gene expression by reducing histone 3 acetylation on lysine 27 [48]. Moreover, acetylation of vestigial-like 4 (VGLL4) at K225 by P300 negatively regulates its binding to the transcription factor TEA Domain Transcription Factor 1 (TEAD1), leading to decreased neonatal cardiomyocytes proliferation and cardiomyocytes necrosis [20]. Inhibition of HDAC1 was also described to decrease the proliferation cardiomyocyte in zebrafish [49]. Indeed, mutation in *HDAC1* gene that induces protein instability is associated with decreased cardiomyocyte proliferation, suggesting an important role of HDAC1 during heart growth [49].

In addition, sirtuins are well-described regulators of aging and have been associated with longevity [50,51,52]. Indeed, mice deficient for *SIRT1* [53,54], *SIRT3* [55] or *SIRT6* [56] display a shorter life span with severe cardiac damage, such as hypertrophy and fibrosis. SIRT1 is well known to be cardioprotective by decreasing oxidative stress and inflammation, which promotes cardiomyocyte survival [54]. Moreover, SIRT3 is required for cardiomyocyte survival under several stress conditions such as serum starvation, genotoxic and oxidative stress [57]. Altogether, these studies demonstrate that KDACs and KATs exert crucial functions during heart development and cardiac ageing.

#### 2.2.2. Cardiac Contraction

Recently, reversible acetylation of sarcomeric proteins has been described as a mechanism regulating cardiac function. The comparison of lysine acetylation patterns from rats as well as from human skeletal muscle biopsies revealed that 80% of the proteins involved in muscle contraction were acetylated [9]. Moreover, proteomic analyses suggested that HDAC6 is localized in Z-disks and acts as a sarcomeric protein desacetylase [58]. Among them, acetylation can impact the β-myosin heavy chain (lysine 34, lysine 58, lysine 213, lysine 429, lysine 951 and lysine 1195) [31], titin [33], CapZβ1 (lysine 199) [34] or cardiac troponin I [35], directly affecting cardiac contraction.

#### 2.2.3. Role of Lysine Acetylation in Cardiac Energy Metabolism and Mitochondrial Activity

Proteomic analyses suggested that the subcellular distribution of lysine-acetylated proteins is tissue dependent [9]. For example, the heart and muscles, both high energy-demanding organs, are tissues with the largest fraction of mitochondrial protein acetylation [9]. Calorie restriction or changes in nutrition specifically affect the mitochondrial acetylome, but not the cytosol or nucleus [59], suggesting different roles for lysine acetylation in the mitochondria, nucleus and cytoplasm. Indeed, mitochondrial lysine acetylation was described to modulate cell metabolism by regulating fatty acid β-oxidation, the tricarboxylic acid cycle, the urea cycle, and oxidative phosphorylation [60].

In the early newborn period, an important change in myocardial energy substrate metabolism occurs with an increase in fatty acid β-oxidation [47], associated with an increase in long chain acyl CoA dehydrogenase (LCAD) and L-3-hydroxy acyl-CoA dehydrogenase (β-HAD) acetylation and activation [12]. At the cardiac level, mitochondrial general control of amino acid synthesis 5 like-1 (GCN5L1) triggers LCAD and β-HAD acetylation [10,11] whereas SIRT3 deacetylates both LCAD and β-HAD [13,14]. The implication of GCN5L1 in cardiac metabolism was confirmed by genetic deletion approaches. Indeed, knockdown of *GCN5L1* in H9c2 cardiomyoblasts decreases maximal mitochondrial respiration and activity of proteins involved in fatty acid oxidation [12].

However, there is no consensus about the effects of lysine acetylation on fatty acid β-oxidation. Indeed, some studies suggested an inhibitory effect [43] whereas others suggested a stimulatory effect [13,14]. As an example, it was reported that an impaired fatty acid β-oxidation was linked to a reduced acetylation and activity of LCAD in the liver [13] or heart [14] of SIRT3 knockout mice. On the other hand, cardiac mitochondrial protein hyperacetylation induced by *SIRT3* deletion is associated with an increased fatty acid β-oxidation rates [43]. This discrepancy could be due to organ, physio(patho)logical status (e.g., diabetes, obesity or newborn period) or acetylated proteins themselves, and are well described in the following review [43].

The implication of sirtuins in metabolic regulation is well detailed in a recent review [61]. Indeed, SIRT6 was described as a gatekeeper of glucose metabolism in cardiomyocytes. SIRT6 partial depletion decreased mitochondrial respiration whereas SIRT6 overexpression enhanced basal oxygen consumption [62]. Conversely, fatty acid uptake increases in *SIRT6*-deficient cardiomyocytes and is decreased in SIRT6-overexpressed cardiomyocytes [63]. Of note, proteins involved in mitochondrial biogenesis can also be acetylated, such as optic atrophy 1 (OPA1) through SIRT3 which is able to activate OPA1 by deacetylation of the residues lysine 926 and lysine 931 [15]. Finally, a cross-talk between SIRT1, peroxisome proliferator-activated receptor gamma coactivator 1 alpha (PGC-1α) and AMP-activated protein kinase (AMPK) has been described to regulate cardiac metabolism [54]. Indeed, PGC-1α is a transcription factor playing a central role in the regulation of cellular energy metabolism, mitochondrial biogenesis and oxidative phosphorylation. SIRT1 could interact with PGC-1α and increase its expression levels [16]. Moreover, AMPK increases cellular NAD^+^ levels and subsequently increases SIRT1 activity, resulting in PGC-1α deacetylation and activation [64].

#### 2.2.4. Mitochondrial Oxidative Stress

It is well described that several mitochondrial proteins involved in oxidative stress are acetylated in the heart [60]. Oxidative stress is defined by a production of reactive oxygen species (ROS) higher than anti-oxidant capacities. Indeed, although different molecular processes may contribute to global oxidative stress, the majority of ROS originates from the mitochondrial compartment in the heart. Excessive ROS production occurs during mitochondrial dysfunction and induces irreversible damage to mitochondria, defining them as significant contributors to the development of cardiovascular disease [65,66]. In this context, we recently described that SIRT3 deacetylates and activates the mitochondrial superoxide dismutase 2 (SOD2), but there is a decreased interaction between SIRT3 and SOD2 induced SOD2 acetylation on lysine 68 and its subsequent inactivation, leading to mitochondrial oxidative stress and dysfunction in hypertrophied cardiomyocytes [27]. Moreover, SIRT3 inhibition increased oxidative stress in neonatal rat cardiomyocytes [27] or human aortic endothelial cells [26].

Other anti-oxidant enzymes were described to be acetylated in the heart, such as peroxiredoxin 1 [28] in the mitochondria or nuclear factor erythroid 2-related factor 2 [29] in the nucleus.

## 3. Implication of Cardiac Acetylation in Metabolic Heart Disease

Post-translational acetylation triggers modification of the activity of several proteins that occurs in obesity, diabetes and early stage of heart failure as detailed below [67,68]. Table 2 summarizes the cardiac modulation of KATs and KDACs expression levels reported in metabolic heart disease.

### 3.1. Cardiac Hypertrophy

Cardiac hypertrophy is a consequence of genetic, mechanic or neurohormonal changes contributing to heart failure progression. Interestingly, total KAT activity was found to be increased in the hearts of mice subjected to phenylephrine (PE), an hypertrophic stimulus [23]. In this context, the increased expression of P300 and KAT2B acetylases may explain hypertrophied cardiomyocytes under PE [23,25,46,69]. In addition, PE-induced hypertrophy also decreases SIRT6 expression, leading to an increase in histone H3 acetylation on lysine 9 (H3K9) [25,46,69]. Interestingly, PE-induced hypertrophy is decreased by treatment with polyunsaturated fatty acid [69] or anacardic acid [23] in association with a decreased H3K9 acetylation. In parallel, overexpression of P300 induces an increased size of cardiomyocytes and a modification of their myofibrillar organization [69], whereas inhibition of P300 by siRNA attenuates hypertrophic response in neonatal rat cardiomyocytes [46]. On the other hand, overexpression of SIRT6 [46] or SIRT3 [27] decreased cardiomyocytes hypertrophy. As an example, NAD inhibits oxidative stress and hypertrophy induced by PE in cardiomyocytes or by angiotensin II in mice [36]. Indeed, NAD requires SIRT3 to deacetylate and activate liver kinase B1 (LKB1) and its target AMPK [36]. Furthermore, a decreased SIRT2 protein expression was reported in hypertrophic hearts from mice whereas cardiac-specific SIRT2 overexpression protected the hearts against angiotensin II–induced hypertrophy [37]. As shown for SIRT3, SIRT2 deacetylates LKB1 at lysine 48, these promote the phosphorylation of LKB1 and the subsequent activation of AMPK signaling [37]. Moreover, SIRT7 levels increase in myocardial tissues after pressure overload induced by transverse aortic constriction in mice [21]. Cardiomyocyte-specific deletion of *SIRT7* exacerbates hypertrophy and fibrosis induced by transverse aortic constriction, suggesting an antihypertrophic role of SIRT7 [21]. Indeed, SIRT7 deacetylates the hypertrophy response transcription factors such as GATA-binding factor 4 (GATA4) in cardiomyocytes [21]. Conversely, P300 acetylates GATA4 and activates its DNA binding activity, inducing cardiac hypertrophy and heart failure [22].

Finally, HDAC3 expression and activity increase during hypertrophy [80] whereas a decreased binding of HDAC3 to myofibrils is observed in PE-induced hypertrophy in neonatal rat cardiomyocytes that increases the acetylation of CapZβ1 at lysine 199 and alters myofibril growth during cardiac hypertrophy [34].

### 3.2. Cardiac Fibrosis

Cardiac fibrosis, defined as an excessive deposition of extracellular matrix in the cardiac muscle, is an important contributor to several heart diseases. Interestingly, in isoproterenol-injected mice, KAT2B activity was found to be increased only in cardiac fibroblasts, whereas no effects were observed in cardiomyocytes isolated from these mice [39]. In this model, KAT2B induces acetylation of SMAD2 and, subsequently, activates SMAD2 and the transforming growth factor β signaling pathway [39]. SMAD3 could also be acetylated during cardiac fibrosis [41,42]. Activation of SIRT1 by resveratrol [41], nicotinamide mononucleotide [42] or geniposide [70] could decrease SMAD3 acetylation and fibrosis. Moreover, inhibition of P300 appears to exert anti-fibrotic functions [82].

Decreased α-tubulin acetylation and increased HDAC6 were also described in cardiac fibroblasts or the heart from isoproterenol-induced fibrosis. Interestingly, treatment of cardiac fibroblasts with tubastatin A, an HDAC6 inhibitor, restored α-tubulin acetylation and decreased fibrosis [83].

### 3.3. Heart Failure

Cardiac hypertrophy and fibrosis are both involved in heart failure development by activation of several intracellular signaling pathways, leading to left ventricular remodeling with systolic dysfunction. Despite the best modern therapeutic management, left ventricular remodeling remains independently associated with heart failure and cardiovascular death at long-term follow-up after myocardial infarction [84]. Moreover, alterations in cardiac energy metabolism, both in terms of changes in energy substrate preference and decreased mitochondrial oxidative metabolism and ATP production, are key contributors to heart failure development [5].

An increase in P300 expression is observed during ischemia/reperfusion injury [85] and myocardial infarction [69]. Interestingly, it was described that activation of P300 acetylase, leading to an increase in acetylated H3K9 induced by myocardial infarction, is reversed by treatment with polyunsaturated fatty acid [69]. Indeed, eicosapentaenoic acid and docosahexaenoic acid directly blocks the histone acetyltransferase activity of P300 and subsequently reduces both hypertrophy and fibrosis induced by myocardial infarction in rats [69]. Other P300 inhibitors were also described to be cardioprotective, such as *Ecklonia stolonifera* Okamura extract [86], curcumin [87] and metformin [88]. Moreover, KAT2B levels are also increased by ischemia/reperfusion injury [73].

The expression and activity of sirtuins are also highly impacted during heart failure. As an example, a decrease in SIRT3 and SIRT6 expressions were observed in human failing hearts that induces a global increase in protein acetylation [63,77]. Proteomic analysis also identified an increased acetylation of mitochondrial proteins induced by transverse aortic constriction [89].

Inhibition of SIRT3 was also described to induce mitochondrial oxidative stress and hypertrophy, notably by increasing the inactive form of SOD2 (acetylated on lysine 68) in hypertrophy [27] or following ischemia/reperfusion [90]. Moreover, deletion of SIRT2 exacerbates cardiac hypertrophy and fibrosis and decreases cardiac ejection fraction and fractional shortening in angiotensin II-infused mice by inhibition of AMPK activation, whereas cardiac-specific SIRT2 overexpression reversed this phenotype [37]. SIRT1 expression and activity are decreased in the heart following ischemia/reperfusion [30]. SIRT1 inhibition was also reported to increase endoplasmic reticulum stress-induced cardiac injury by decreasing eukaryotic initiation factor 2 alpha (eIF2α) deacetylation on lysine 143 in cardiomyocytes and in adult-inducible SIRT1 knockout mice [91].

Acetylation of contractile proteins is also modified during heart failure. For example, acetylation of β-MHC on lysine 951 was decreased in both ischemic and non-ischemic failing hearts [31].

### 3.4. Obesity

Obesity is a major cause of disability and is often associated with cardiac hypertrophy, fibrosis, type 2 diabetes, obstructive sleep apnea and alteration of cardiac metabolism. It has notably described an increase in genes involved in fatty acid oxidation [11]. Moreover, several studies have suggested an increase in acetylation raised from the non-enzymatic reaction of high levels of acetyl-CoA generated during a high-fat diet (HFD) and obesity [5]. Furthermore, hyperacetylation of mitochondrial proteins and metabolic inflexibility was reported in response to HFD or obesity. SIRT3 expression is decreased in left ventricles of obese patients and is associated with an increased level of brain natriuretic peptide (BNP), a marker of cardiac dysfunction and of protein acetylation [17]. It was also reported that exposure of cardiomyoblasts H9c2 to palmitate led to an increase in both SIRT3 and GCN5L1 RNA levels [11]. At the metabolic level, HFD induced an increase in SCAD and LCAD acetylation and activity [11]. Moreover, the acetylation level of α-tubulin on lysine 40 is increased in the hearts of HFD mice and a pharmacological activation of α-tubulin acetylation decreases glucose transport [92].

In the other hand, cardiac specific inhibition of SIRT6 in mice exposed to HFD-induced cardiac hypertrophy and lipid accumulation [93]. In this context, SIRT6 activated the expression of endonuclease G and SOD2, that could decrease oxidative stress and hypertrophy [93].

### 3.5. Type 2 Diabetes

As obesity, type 2 diabetes is also associated with several cardiac dysfunctions such as cardiac hypertrophy and fibrosis, and acetylation may be involved in these mechanisms. Indeed, SIRT1 expression and activity are decreased in the heart of diabetic rats, induced by HFD and streptozotocin injection and, conversely, an up-regulation of SIRT1 by adenovirus attenuates cardiac dysfunction and oxidative stress [30]. Moreover, in a model of sucrose-fed rats, cardiac dysfunction is associated with a decreased SIRT3 protein expression (despite an increase at RNA level) and an increased GCN5L1 (protein and RNA) and mitochondrial protein acetylation [11,17]. Decreased SIRT3 levels are also associated with an increased acetylation of SOD2, and an increased oxidative stress and apoptosis in heart of diabetic mice [79]. Interestingly, exposure of cardiomyoblasts H9c2 to high glucose concentration decreased SIRT3 [78] and increased GCN5L1, oxidative stress and autophagy mediated by cytoplasmic Forkhead box O1 (FOXO1) acetylation [11,76]. Conversely, expression of GCN5L1 is decreased in hearts of diabetic ZSF1 rats, a model characterized by obese animals with hyperglycemia, hyperinsulinemia and cardiac dysfunction [10]. This decreased GCN5L1 expression and activity, with no modulation of SIRT3, induced a decrease of short- and long-chain acyl CoA dehydrogenases and a reduced respiratory capacity [10]. One explanation could be the transition from prediabetes, in which GCN5L1 becomes elevated to promote mitochondrial fatty acids oxidation, to an overt diabetic state, in which GCN5L1 expression is downregulated leading to an overall decrease in mitochondrial fuel oxidation [10].

On the other hand, OVE26 mice, a mouse model of type 1 diabetes, develop cardiac dysfunction with increased left ventricle diameters and fibrosis and decreased ejection fraction associated with an increase in HDAC3 activity, oxidative stress and inflammation [81]. Interestingly, the selective HDAC3 inhibitor RGFP966 reversed the cardiac phenotype in these mice [81]. In another model of type 1 diabetes, injection of streptozotocin in rats induced cardiac dysfunction, cardiac hypertrophy, fibrosis and inflammation [72] as well as oxidative stress [28]. This oxidative stress is due to an increased HDAC6 activity leading to a decrease in the acetylated form of peroxiredoxin 1, suggesting that acetylation of peroxiredoxin 1 is involved in this activation [28]. In this context, tubastatin A, a highly selective inhibitor of HADC6, may represent an interesting pharmacological target since treatment of diabetic rats with tubastatin A increased acetylation of peroxiredoxin 1 and decreased oxidative stress and cardiac dysfunction [28]. Conversely, SIRT1 expression and activity appear to be decreased in the left ventricles of streptozotocin-induced diabetic rats [72] and mice [38]. In these models, reduced SIRT1 expression is associated with acetylation of endothelial nitric oxide synthase [72] or Akt [38]. In addition, SIRT6 expression is also decreased in the hearts of diabetic mice [63], suggesting that sirtuin family members may play a crucial role in type 1 diabetes-associated cardiac dysfunctions.

### 3.6. Pharmacological Modulation of Cardiac Acetylation

Due to the major implication of KATs and KDACs in cardiac dysfunction induced by obesity, type 2 diabetes or heart failure, targeting these enzymes could be beneficial for patients.

First, inhibition of P300 appears to be a promising approach to inhibit cardiac hypertrophy and fibrosis induced by several stimuli. As an example, oral administration of eicosapentaenoic acid (1 g/kg) or docosahexaenoic acid (1 g/kg) one week after myocardial infarction preserved fractional shortening and decreased cardiac hypertrophy and perivascular fibrosis [68]. *Ecklonia stolonifera* Okamura extract, an algae traditionally used in Japanese foods, inhibits PE-induced hypertrophy in neonatal rat cardiomyocytes and restores fractional shortening and reduced cardiac hypertrophy and fibrosis by oral administration one week after myocardial infarction in rats [73]. Oral administration of curcumin (50 mg/kg/day) acts also as a P300 inhibitor and decreases posterior wall thickness, cardiac hypertrophy and perivascular fibrosis induced by hypertension [74]. Finally, metformin, that directly inhibits P300-mediated acetylation of H3K9, blocks PE-induced cardiomyocytes hypertrophy [75].

Based on the cardioprotective effects of SIRT3, it seems essential to develop new therapeutic strategy to increase its expression or activity. Recent reviews have well described these molecules [94,95]. For example, exogenous NAD is able to inhibit hypertrophy induced by PE in cardiomyocytes or by angiotensin II in mice [36]. Resveratrol, an activator of both SIRT1 and SIRT3, is also described to be cardioprotective [41].

## 4. Conclusions

Despite available therapies, cardiovascular diseases, and particularly ischemic diseases, still remain the first cause of mortality and morbidity in the world. Obesity and type 2 diabetes are some of the major risk factors of cardiovascular diseases. Acetylation is an essential mechanism involved in several processes contributing to cardiac diseases such as cell metabolism, gene transcription or enzymatic activity. In this context, targeting enzymes responsible of acetylation/deacetylation could be beneficial and is a very promising research area, as demonstrated by the development of inhibitor targeting P300 [82], modulators of SIRT6 [96] or agonists of SIRT3 [94,95] to treat cardiac dysfunctions.

## Figures and Tables

**Figure 1 biomedicines-10-01834-f001:**
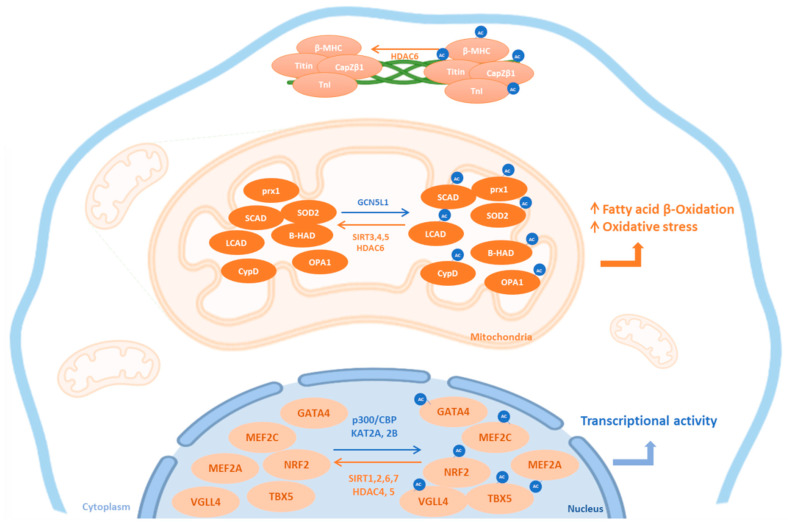
Subcellular localization of cardiac KATs and KDACs. This figure summarizes the subcellular localization of the most common lysine acetyltransferases (KATs, blue) and lysine deacetylases (KDAC, orange) and their cardiac targets. Ac: acetylated form; HDAC: histone deacetylase; SIRT: sirtuins; β-MHC: beta-myosin heavy chain; TnI: Troponin I; prx1: peroxiredoxin 1; LCAD: long chain acyl CoA dehydrogenase; SCAD: short chain acyl CoA dehydrogenase; β-HAD: L-3-hydroxy acyl-CoA dehydrogenase; SOD2: superoxide dismutase 2; CypD: Cyclophilin D; OPA1: optic atrophic 1; TBX5: T-Box transcription factor 5; VGLL4: vestigial-like 4; GATA4: GATA-binding factor 4; MEF: myocyte enhancer factor; Nrf2: nuclear factor erythroid-2-related factor 2; GCN5L1: general control of amino acid synthesis 5 like-1; CBP: CREB binding protein.

**Figure 2 biomedicines-10-01834-f002:**
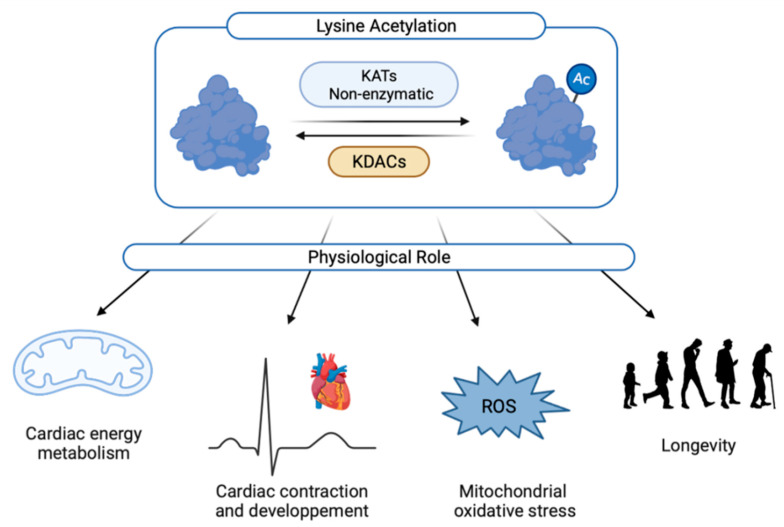
Physiological roles of cardiac acetylation. Ac: acetylated form; KATs: lysine acetyltransferases; KDACs: lysine deacetylase; ROS: reactive oxygen species.

**Table 1 biomedicines-10-01834-t001:** Most common proteins modified by cardiac acetylation in cardiovascular and metabolic diseases.

Class	Name	KAT	KDAC	Function of Acetylation in the Heart	References
Mitochondrial proteins	LCAD/SCAD	GCN5L1	SIRT3	Increased activity and modulated fatty acid oxidation	[5,10,11,12,13]
	β-HAD	GCN5L1	SIRT3	Increased activity and modulated fatty acid oxidation	[5,12,14]
	OPA1	Unknown	SIRT3	Decreases its activity	[15]
	PGC1α	Unknown	SIRT1	Increases its expression	[16]
	Cyclophilin D	GCN5L1	SIRT3	Induces mPTP opening	[17]
Transcription factors	TBX5	KAT2A, KAT2B	HDAC4 HDAC5	Increases transcriptional activity	[18,19]
	VGLL4	P300	Unknown	Negatively regulates its binding to TEAD1	[20]
	GATA4	P300	SIRT7	Activates its DNA binding activity	[21,22]
	MEF2A	P300 KAT2B	HDAC5	Increased hypertrophy	[23,24]
	MEF2C	KAT2B	HDAC5	Increased hypertrophy	[24,25]
Anti-oxidant proteins	SOD2	Unknown		Decreases SOD2 activity	[26]
			SIRT3	Increased mitochondrial oxidative stress and hypertrophy	[27]
	Prx1	Unknown	HDAC6	Increased peroxide-reduction activity	[28]
	Nrf2	Unknown	SIRT1	Decreases its activity	[29]
	eNOS	Unknown	SIRT1	Inactive form	[30]
Contractile proteins	β-MHC	Unknown	HDAC6	Impact myosin head positioning	[31,32]
	Titin	Unknown	HDAC6	Cardiac contraction	[33]
	CapZβ1	Unknown	HDAC3/6	Cardiac contraction	[34]
	TnI	Unknown	HDAC6	Cardiac contraction	[35]
Signaling pathway	LKB1	Unknown	SIRT2 SIRT3	Induces its activation by phosphorylation	[36,37]
	Akt	Unknown	SIRT1	Inhibition of Akt phosphorylation and activation	[38]
	SMAD2	KAT2B	SIRT1	Induced fibrosis	[39,40]
	SMAD3	Unknown	SIRT1	Induced fibrosis	[40,41,42]

KAT: lysine acetyltransferase; KDAC: lysine deacetylase; LCAD: long chain acyl CoA dehydrogenase; SCAD: short chain acyl CoA dehydrogenase; β-HAD: L-3-hydroxy acyl-CoA dehydrogenase; OPA1: optic atrophic 1; PGC-1α: peroxisome proliferator-activated receptor-gamma coactivator; TBX5: T-Box transcription factor 5; VGLL4: vestigial-like 4; TEAD1: TEA Domain Transcription Factor 1; GATA4: GATA-binding factor 4; MEF: myocyte enhancer factor; SOD2: superoxide dismutase 2; Prx1: peroxiredoxin 1; Nrf2: nuclear factor erythroid-2-related factor 2; eNOS: endothelial nitric oxide synthase; β-MHC: beta-myosin heavy chain; TnI: troponin I; LKB1: liver kinase B1; GCN5L1: general control of amino acid synthesis 5 like-1; SIRT: sirtuins; HDAC: histone deacetylase; mPTP: mitochondrial permeability transition pore.

**Table 2 biomedicines-10-01834-t002:** Modulation of KATs and KDACs expression in heart failure and metabolic diseases.

KATs			KDACs		
Name	Heart Failure	Metabolic Diseases	Name	Heart Failure	Metabolic Diseases
P300	Increase [46,69]	Unknown	SIRT1	Decrease [70,71]	Decrease [30,38,72]
KAT2B	Increase [23,25,73]	Unknown	SIRT2	Decrease [37]	Decrease [74,75]
GCN5L1	Unknown	Increase in pre-diabetes [11,76]	SIRT3	Decrease [77]	Decrease [17,78,79]
		Decrease in diabetes [10]	SIRT6	Decrease [46,63]	Decrease [63]
			SIRT7	Increase [21]	Unknown
			HDAC3	Increase [80]	Increase [81]
			HDAC6	Unknown	Increase [28]

KATs: lysine acetyltransferases; KDACs: lysine deacetylases; SIRT: sirtuins; HDAC: histone deacetylase; GCN5L1: general control of amino acid synthesis 5 like-1.

## Data Availability

Data sharing not applicable.

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
