# Peer review of "Cardiac Acetylation in Metabolic Diseases"

_biomedicines, 2022, doi:10.3390/biomedicines10081834_

Round 1

Reviewer 1 Report

Thank you very much for the opportunity to review this manuscript.

Dubois-Deruy and colleagues report the key roles of lysine acetylation in physiological conditions as well as in cardiovascular disease (CVD) and CVD-associated disorders, with particular emphasis on the enzymes involved and the main targets. The review is interesting and well written. Therefore, I have no major concerns about the publication of this paper.

Although the manuscript is well written, "minor revision" is required to correct the few english typos. e.g. raw 229: "induing" has to be changed with "inducing"

raw 295: "decreased" has to be changed with "decrease" 

Best regards

Author Response

Thank you very much for the opportunity to review this manuscript. Dubois-Deruy and colleagues report the key roles of lysine acetylation in physiological conditions as well as in cardiovascular disease (CVD) and CVD-associated disorders, with particular emphasis on the enzymes involved and the main targets. The review is interesting and well written. Therefore, I have no major concerns about the publication of this paper.

We thank you sincerely for your time to read our manuscript, and we appreciate the positive comments. We revised our manuscript to address your concerns.

Although the manuscript is well written, "minor revision" is required to correct the few english typos. e.g. raw 229: "induing" has to be changed with "inducing", raw 295: "decreased" has to be changed with "decrease"

We corrected the typos and checked carefully our revised manuscript to avoid other typos error.

Reviewer 2 Report

The present draft addresses an interesting and pertinent topic. Dubois-Deruy and colleagues analysed the role of lysine acetylation in metabolic diseases. 

The manuscript is well and carefully written and gives a lot of information on the cardiac acetylation, regulation and function as well as the contribute to heart failure. 

Minor aspects could be improved. I believe that this manuscript could be considered for publication.

- Figure 1:  In the figure legend, please specify the correlation between KATs KDACs and the colour (blue and orange)

- Figure 2:  In the figure legend, “ROS” should be given in full.

- Table 1:  In the table, please specify the references. 

- Some studies suggesting an inhibitory effect of lysine acetylation on fatty acid β-oxidation, while others suggesting a stimulatory effectPlease elaborate on this.

- In the 3.3. Heart failure section, mitochondrial enzyme acetylation in cardiac ischemia-reperfusion injury should be considered.

Author Response

The present draft addresses an interesting and pertinent topic. Dubois-Deruy and colleagues analysed the role of lysine acetylation in metabolic diseases. The manuscript is well and carefully written and gives a lot of information on the cardiac acetylation, regulation and function as well as the contribute to heart failure. Minor aspects could be improved. I believe that this manuscript could be considered for publication.

We thank you sincerely for your time to read our manuscript and for formulating the following comments. We appreciate the positive comments and we revised our manuscript to address your concerns.

- Figure 1:  In the figure legend, please specify the correlation between KATs / KDACs and the colour (blue and orange)

We improve our figure 1 legend by adding the correlation between KATs / KDACs and the colour as following:

Lines 241-249 “Figure 1. Subcellular localization of cardiac KATs and KDACs. This figure summarizes the subcellular localization of the most common lysine acetyltransferases (KATs, blue) and lysine deacetylases (KDAC, orange) and their cardiac targets.”

- Figure 2:  In the figure legend, “ROS” should be given in full.

We added ROS signification in figure 2 legend.

Line 310 “Figure 2. Physiological roles of cardiac acetylation. Ac : acetylated form; KATs: lysine acetyltransferases; KDACs: lysine deacetylase; ROS: reactive oxygen species.”

- Table 1:  In the table, please specify the references.

We specified references in both Tables 1 and 2.

- Some studies suggesting an inhibitory effect of lysine acetylation on fatty acid β-oxidation, while others suggesting a stimulatory effect. Please elaborate on this.

We thank the reviewer for suggesting this effect and we detailed the different studies in the manuscript as following:

Lines 363 – 372 “However, there is no consensus about the effects of lysine acetylation on fatty acid β-oxidation. Indeed, some studies suggested an inhibitory effect [43] whereas others suggested a stimulatory effect [13,14]. As example, it was reported an impaired fatty acid β-oxidation linked to a reduced acetylation and activity of LCAD in the liver [13] or heart [14] of SIRT3 knockout mice. On the other hand, cardiac mitochondrial protein hyperacetylation induced by SIRT3 deletion is associated with an increased fatty acid β-oxidation rates [43]. This discrepancy could be due to organ, physio(patho)logical status (e.g. diabetes, obesity, or newborn period) or acetylated proteins themselves and are well described in the following review [43].”
